# Double-Layered Films Based on Furcellaran, Chitosan, and Gelatin Hydrolysates Enriched with AgNPs in Yerba Mate Extract, Montmorillonite, and Curcumin with Rosemary Essential Oil

**DOI:** 10.3390/polym14204283

**Published:** 2022-10-12

**Authors:** Ewelina Jamróz, Magdalena Janik, Luís Marangoni, Roniérik Pioli Vieira, Joanna Tkaczewska, Agnieszka Kawecka, Michał Szuwarzyński, Tomasz Mazur, Joanna Maria Jasińska, Paweł Krzyściak, Lesław Juszczak

**Affiliations:** 1Department of Chemistry, University of Agriculture, ul. Balicka 122, PL-30-149 Kraków, Poland; 2Packaging Technology Center, Institute of Food Technology, Campinas 13083-862, Brazil; 3Department of Bioprocess and Materials Engineering, School of Chemical Engineering, University of Campinas, Campinas 13083-862, Brazil; 4Department of Animal Product Technology, Faculty of Food Technology, University of Agriculture, ul. Balicka 122, PL-30-149 Kraków, Poland; 5Department of Product Packaging, Cracow University of Economics, ul. Rakowicka 27, PL-31-510 Kraków, Poland; 6Academic Centre for Materials and Nanotechnology, AGH University of Science and Technology, Al. Mickiewicza 30, PL-30-059 Kraków, Poland; 7Department of Infection Control and Mycology, Faculty of Medicine, Jagiellonian University Medical College, ul. Czysta 18, PL-31-121 Kraków, Poland; 8Department of Dietetics and Food Studies, Faculty of Science and Technology, Jan Długosz University in Częstochowa, ul. Armii Krajowej 13/15, PL-42-200 Częstochowa, Poland

**Keywords:** biopolymer, bioactive compounds, active films

## Abstract

Double-layered active films based on furcellaran (1st layer—FUR), chitosan, and gelatin hydrolysates (2nd layer—CHIT+HGEL) were successfully prepared. Bioactive ingredients were added to the 1st film layer: AgNPs, which were synthesized in situ with yerba mate extract; montmorillonite clay (MMT); and different loads of ethanolic curcumin (CUR) extract enriched with rosemary essential oil (REO). SEM images confirmed the presence of AgNPs with a size distribution of 94.96 ± 3.33 nm throughout the films, and AFM and SEM photos indicated that the higher substance concentrations had rougher and more porous film microstructures. However, the water vapor transmission rate was reduced only at the lowest load of this ingredient. Despite the tensile strength of the films having decreased, the incorporation of the compounds showed a tendency towards reducing the modulus of elasticity, resulting in a lower stiffness of the composites. The addition of CUR and AgNPs improved the UV light barrier properties of the materials. The presented films showed quick reactions to changes in the pH value (from orange to red along with an increase in pH from 2 to 10), which indicates their potential use as indicators for monitoring the freshness of food products. Composite No. 2 showed the highest antimicrobial potential, while none of the presented films showed an antifungal effect. Finally, the antioxidant activities of the films increased dramatically at higher AgNP and CUR loads, suggesting an outstanding potential for active food packaging applications.

## 1. Introduction

Research aimed at the production of films based on biopolymers is increasing in the scientific community, mainly because they are of renewable and biodegradable origins. The main biopolymers used in the manufacturing of films for food packaging are based on proteins and polysaccharides of animal and plant origin, e.g., furcellaran, chitosan, and hydrolyzed gelatin [1,2,3].

Furcellaran (FUR) is a naturally sulfated anionic polysaccharide obtained from red algae extract (*Furcellaria lumbricalis*). FUR is traditionally found in nature as a mixture of sodium, potassium, magnesium, and calcium salts of a linear polymer composed of [→4)-3,6-anhydro-d-galactopyranose-(1→3)-galactopyranose-4′- sulfate -(1→] structural units [1,4]. This polysaccharide has been investigated for the production of films, mainly because it is nontoxic, biodegradable, biocompatible, and water soluble and has an exceptional ability to form gels [5]. However, FUR films are highly soluble and have poor mechanical properties [5,6].

Chitosan (CHIT) is an abundant polycationic polysaccharide obtained from the deacetylation of chitin, which is mainly found in the exoskeletons of crustaceans [7]. It consists of two subunits, d-glucosamine and N-acetyl-d-glucosamine, linked linearly by 1,4-glycosidic bonds [8]. CHIT is also used for the production of active packaging films, mainly for its intrinsic antimicrobial properties, biodegradability, and biocompatibility. Although it is possible to obtain CHIT films with relatively good tensile strength, they are often fragile (i.e., with low ductility), which limits their applications [9].

Hydrolysed gelatin (HGEL) is a protein extracted from collagen found in industrial by-products from the fishing industry [10] and can easily be prepared as a film [11]. However, their mechanical properties are poor, and they are highly vulnerable when exposed to environments with high humidity, which is an obstacle for their practical application as a packaging film [12].

Although these renewable polymeric matrices are being widely studied, it is observed that their individual uses present numerous limitations for practical applications as food packaging. The main disadvantages of these materials are related to their processing limitations as well as their poor mechanical and barrier properties and low water resistance [13]. A potential solution to these problems could be the development of blends and/or double-layered films [6]. The main advantage of the second strategy is the increase in the barriers for preventing gas permeation, in addition to providing each layer with specific properties [6,14]. There are studies that have investigated the physical and chemical properties of multilayer films based on biopolymers [15,16,17]. Luciano et al. [18] examined double-layered films, based on gelatin in both layers, in which the nisin and/or “pitanga” leaf hydroethanolic extracts were added in the second layer. Cruces et al. [19] received multilayer films of pectin-beeswax/colophony-pectin. Thanks to this composition, it was possible to reduce the values of water vapor permeability.

In previous works, we developed single-layer films based on chitosan, furcellaran, and gelatin hydrolysate [20,21]. However, the developed films were characterized by excessive water vapor permeability, low parameters of mechanical properties, and excessive solubility, which limited the presented films in practical use. Therefore, in this study, it was decided to use the above-mentioned components as a matrix for the preparation of double-layered films. Due to the poor functional properties of furcellaran films, it was decided to enrich the furcellaran-based layer with several active ingredients in the double-layered films. The addition of MMT was intended to improve the functional properties of the furcellaran layer because the lamellar structure of MMT can improve the mechanical, barrier, and moisture-resistance properties of biopolymer-based films [22]. Another strategy implemented to improve film properties is the incorporation of nanoparticles and/or compounds with bioactive attributes. In the present study, bioactive nanoparticles were incorporated into the films. The concept of “green synthesis” was applied in the preparation of silver nanoparticles (AgNPs) by employing yerba mate extract (YM). The green synthesis method involves a single economical and environmentally friendly bioreduction step that can easily be reproduced on a large scale. A series of biomolecule-rich natural materials can act as NP-reducing agents, such as plant extracts [23,24]. Moreover, the application of AgNPs-YM can synergistically confer distinct properties from AgNPs and YM to the films. For example, AgNPs are well-known for their powerful antimicrobial activities [25], and YM extract is known for possessing strong antioxidant properties [26]. In addition to incorporating AgNPs-YM in the double-layered FUR/CHIT+HGEL films, this study employed an ethanolic curcumin extract (CUR) enriched with rosemary essential oil (CUR-REO) to enhance the intelligent properties of the tested films. CUR is an important phenolic compound extracted from turmeric rhizomes (*Curcuma longa* Linn.). It is often applied to active films for its various beneficial effects, such as its antioxidant and antimicrobial activities [27]. Rosemary essential oil has also been used in the production of active packaging with antioxidant and antimicrobial properties [28], and its bioactive properties are due to two main compounds, 1,8-cineole and camphor [29]. However, in our work, the main purpose of using this ingredient was to impart intelligent properties to the tested films. Thus, the combination of these ingredients in double-layer films means that the obtained materials can potentially be used in the packaging of food products. The combination of these ingredients can eliminate the functional disadvantages of biopolymer films. The use of chitosan, furcellaran, and gelatin hydrolysate as components of the biopolymer matrix also reduces the quantity of unused by-products from the food industry. So far, the interactions between these types of components of a biopolymer matrix and the proposed active ingredients are not known. Therefore, this work introduces new knowledge about multilayer composite films.

Our scientific team is the only one in the world that receives multilayer films based on furcellaran, chitosan, and gelatin hydrolysate. The aim of this study was to develop advanced double-layer films from furcellaran (1st layer–FUR), chitosan, and a gelatin hydrolysate blend (2nd layer–CHIT+HGEL) enriched with bioactive and reinforcing agents in the 1st film layer. The green synthesis method was implemented in the preparation of AgNPs with yerba mate extract. Moreover, different loads of ethanolic curcumin extract with rosemary essential oil (CUR-REO) and montmorillonite (MMT) were also incorporated into the 1st layer to potentialize the properties. The effect of the CUR load on the films was assessed based on its morphology, chemical interactions, and physicochemical and biological properties. In addition, it was examined whether the films can act as potential indicators of the freshness of food products during their storage.

## 2. Materials and Methods

### 2.1. Materials

Furcellaran (FUR) (type 7000) was purchased from Est-Agar AS (Karla village, Estonia). The chemical content composition of FUR (molecular weight (M_w_) = 2.951 × 10^5^) was: carbohydrates 79.61%, protein 1.18%, and fat 0.24%. Chitosan (CHIT) (molecular weight ~890,000, CAS number 9012-76-4, ≥90% deacetylation, 100–300 cP viscosity, particle size ≤ 100 mesh) and curcumin (Cat. No.: A147405; purity 98%; Mw 368.38) were supplied from POL-AURA (Zabrze, Poland). Carp skin gelatin hydrolysates were prepared in accordance with the method described in an earlier study by Tkaczewska, Jamróz, Kulawik, Morawska, and Szczurowska (2019). Rosemary essential oil was purchased from Senti (Olsztyn, Poland). Montmorillonite K30 (Cat. No.: 69904) was procured from Sigma-Aldrich (Poznań, Poland). Yerba mate *Ilex paraguariensis* Taragui Sin Palo (Las Marias, Argentina) was obtained from a local market (Krakow, Poland). TWEEN 80 was purchased from Chempur (Piekary Śląskie, Poland). All of the chemical reagents were applied without being subjected to further purification.

### 2.2. Preparation of Curcumin Extract with Rosemary Essential Oil

First, 120 mg of curcumin was dissolved in 30 mL of ethylic alcohol. Subsequently, the solution prepared in this way was placed in an ultrasonic bath for 20 min (temperature 35 °C). Then, 1 mL of rosemary essential oil and 1 mL of TWEEN 80 were added to the clear solution to stabilize the emulsion.

### 2.3. Preparation of MMT Solution

During this step, 250 mg of MMT was added to 50 mL of a 2% acetic acid solution and stirred on a magnetic stirrer for 6 h.

### 2.4. Preparation of Plant Extracts

Yerba mate leaves (10 g) were added to distilled water (100 mL) and heated to 80 °C for 30 min using a magnetic stirrer. The aqueous extract was filtered three times and cooled down to room temperature (∼25 °C).

### 2.5. Synthesis of Silver Nanoparticles (AgNPs)

The procedure for preparing the AgNPs in the extract was performed according to Jamróz et al. [30], with slight modifications. From the prepared solution, 5 mL of plant extract was taken, and 15 mL of distilled water was added. The green synthesis of AgNPs was performed with the addition of 0.4 mL of AgNO_3_ (0.1 M) to the prepared solution, and the reaction took place after approx. 20 min.

### 2.6. Particle Size and Zeta Potential

The characterization of the nanoparticles was obtained using dynamic light scattering (DLS) (Zetasizer Ultra Red, Malvern Instruments Ltd., Worcestershire, UK).

### 2.7. Double-Layered Film Preparation

First, 1 g of chitosan (CHIT) was dissolved in 100 mL of acetic acid solution (2% *v*/*v*) with stirring for 12 h. A furcellaran film-forming solution (FUR) was prepared by dissolving 1% (*w*/*v*) furcellaran powder in hot water (120 °C) under magnetic stirring for 2 h. For every 100 mL of CHIT solution, 1 g of gelatin hydrolysate (HGEL) was added. Afterward, the glycerol (1 mL per 100 mL of film-forming solution) was added as a plasticizer and mechanically shaken at 60 °C for 1 h.

The control sample consisted of two layers. The first one was a 200 mL solution of FUR, while the second one contained 110 mL of CHIT+HGEL.

Furthermore, 6 mL of CUR-REO, 11 mL of MMT, and 6 mL of AgNPs-YM extracts were added to the FUR solution. The solution prepared in this way was poured onto a tray. When the solution turned into a gel, 110 mL of the CHIT+HGEL solution was poured, which constituted the 2nd layer. This type of film is described as composite No. 1.

The next films were prepared in the same way using 12 mL of CUR-REO, 11 mL of MMT, and 6 mL of AgNPs-YM extract (composite No. 2) or 18 mL of CUR-REO, 11 mL of MMT, and 6 mL of AgNPs-YM extract (composite No. 3).

The films were then dried under a fume hood for 2 days at room temperature.

### 2.8. UV-Vis Spectroscopy Analysis

A UV-Vis analysis (with an absorbance spectrum of 200–800 nm) was performed using a UV-5500 spectrophotometer (UV 5500 Metash, Shanghai, China).

### 2.9. Atomic Force Microscopy (AFM)

Atomic force microscope (AFM) images were obtained with a Dimension Icon XR atomic force microscope (Bruker, Santa Barbara, CA, USA). The parameters of the measurement were described previously (Jamróz, Janik, et al., 2021).

### 2.10. Scanning Electron Microscopy (SEM)

Sample morphologies were analyzed using an SEM/FIB Quanta 3D 200i (FEI, Hillsboro, OR, USA) microscope.

### 2.11. Color Parameter

The surface color of the film was measured, and the method was described previously [30], where the material was evaluated via the reflection method using a Color i5 spectrometer (X-Rite, Grand Rapids MI, USA, illuminant D65). The results are presented as L* (lightness), a* (red–green), and b* (yellow–blue). In addition, the total value of the color difference (ΔE) was calculated according to Equation (1):∆E = √((∆a)^2^ + (∆b)^2^ + (∆L)^2^)(1)
where the ΔE values are the total differences between the color of the CHIT+HGEL/FUR film as a standard and those of the active films with CUR+REO in different concentrations.

### 2.12. Thickness

The film thickness was determined using a Mitotuyo no. 7327 manual instrument (Kawasaki, Japan). Measurements were performed on at least five random locations.

### 2.13. Water Content and Solubility

The water content and solubility of the films were determined according to the methods proposed by Kavoosi et al. [31] and Souza et al. [32], with some modifications. Three randomly selected samples of each type of film (3 cm × 3 cm) were weighed (~0.0001 g) and characterized as the initial weight (W_1_); the samples were then dried in an oven at 70 °C for 24 h to determine the initial dry matter (W_2_). Each film was immersed into 30 mL of Milli-Q water, covered, and stored for 24 h at room temperature (25 °C ± 2 °C). The film samples were removed after 24 h and dried in an oven at 70 °C for 24 h to determine the undissolved final dry weight (W_3_). Three measurements were taken for each film sample to calculate the average values of the parameters. The water content and film solubility were calculated by Equations (2) and (3), respectively: water content = (W_1_ − W_2_/W_1_) × 100%(2)
solubility = (W_2_ − W_3_/W_2_) × 100%(3)

### 2.14. Water Vapor Transmission Rate (WVTR)

The water vapor transmission rate (WVTR) was measured by a method described previously [30]. A glass vessel was filled with silica gel, which was then covered with the film under analysis. After that, it was placed in a regulated microclimatic (condition) chamber at a temperature of 25 °C and 75% of relative humidity. Next, the vessel was weighed at the indicated periods of time. WVTR was assessed on the basis of weight gain. The tested specimens were prepared and then air-conditioned in a normative manner. The WVTR was calculated according to the following formula: WVTR [g/m^2^ × d] = 240 × (weight of water ÷ surface penetration) × 24(4)

### 2.15. Contact Angle Determination

The water contact angle (WCA) measurement method has been described previously [30]. The contact angle, with regard to the water contents of the films under study, was determined using the sessile drop method with a video-based measuring system for assessing the contact angle (OCA, Dataphysics, Filderstadt, Germany). This was performed at room temperature (~23 °C). A droplet (in the amount of 10 µL) of deionized water was gently dripped onto the film surface via a microinjector. Then, the image was obtained. Measurements were carried out for five samples.

### 2.16. Thermal Properties

The thermal properties of the films were measured by a DSC 204F1 Phoenix differential scanning calorimeter (Netzsch, Selb, Germany).

### 2.17. Mechanical Properties

A determination of the tensile properties was conducted on the basis of standards concerning plastics, with the general principles being presented in ISO 527-1. The test conditions were consistent with ISO standard 527-3:1995.

### 2.18. Antioxidant Activity

An antioxidant activity (FRAP assay and DPPH analysis) evaluation was performed according the method described by Jamróz, Tkaczewska, Juszczak, Zimowska, Kawecka, Krzyściak, and Skóra [6]. In order to determine the FRAP, its solution was freshly prepared just before the analysis. The solution comprised an acetate buffer (pH 3.6), a ferric chloride (20 mM), and a 2,4,6-tripyridyl-s-triazine solution (10 mM TPTZ in 40 mM HCl) at a ratio of 10:1:1 (*v*/*v*/*v*), respectively. First, the FRAP solution was incubated in the dark at a temperature of 37 °C for 30 min and was then mixed with a film extract at a ratio of 0.4:3.6 (*v*/*v*). The solution was incubated once more at 37 °C for 10 min in dark conditions, followed by absorbance measurement at 593 nm via a Helios Gamma UV-1601 spectrophotometer (Thermo Fisher Scientific, Waltham, MA, USA). For the DPPH assay, the film extract was mixed with 0.1 mM DPPH·. This was performed in an ethanol solution at a ratio of 0.2:2.8 (*v*/*v*). Next, the mixture was incubated in the dark for a period of 30 min. Afterwards, the solution absorbance was evaluated at 517 nm (Helios Gamma, Thermo Fischer Scientific, USA) and compared with the blanks in which the film extract had been replaced with distilled water. The results are expressed as the % of inhibition. The analyses were performed after 30 min in duplicate for three samples of the films (n = 2 × 3).

### 2.19. Antimicrobial Activity

The antimicrobial effects against bacteria (*Staphylococcus aureus* ATCC 29213, *Escherichia coli* ATCC 25922, *Pseudomonas aeruginosa* ATCC 9027, *Enterococcus faecalis* ATCC 29212, and *Salmonella enterica* ATCC BAA 664) and fungi (*Candida krusei* ATCC 6258, *Candida albicans* ATCC 90028, *Aspergillus niger* ATCC 16 404, and *Aspergillus flavus* ATCC 204 304) were evaluated using an agar disc diffusion assay. The films were cut into pieces (1 cm × 1 cm) and placed on agar plates (Muller Hinton 2 in the case of bacteria and Sabouraud Glucose Agar in the case of yeast). Then, each 10 mL of the liquefied (40–50 °C) medium was inoculated with a standard microorganism suspension (100 μL of 0.5 McFahrland inoculum per 10 mL; the final concentrations of bacteria or yeast in the media that were poured onto Petri dishes were 1–5 × 10^6^ CFU mL^−1^ and 1–5 × 10^4^ CFU mL^−1^, respectively). The obtained media were further poured onto agar plates with the tested films and were then allowed to solidify. Then, they were incubated at a temperature of 37 °C for a duration of 24 h. The measuring was performed visually, comparing the growth of microorganisms in the areas under and around the pieces of film.

### 2.20. pH-Responsive Color Changes

The pH-responsive color changes of the tested films were evaluated following the method of Ezati and Rhim [33], with modifications. The tested films were immersed in solutions of alkaline and acidic pH, and changes in the colors of the films were observed. The obtained results were visualized using a camera.

### 2.21. Statistical Analysis

In order to assess the significance of the differences between the average values, the data were analyzed using a one-way analysis of variance. The significance of the differences between the means was established via Fisher’s LSD test; the assumed level of statistical significance was 0.05. Calculations were carried out via Statistica 12.0 (StatSoft Inc., Tulsa, OK, USA).

## 3. Results and Discussion

### 3.1. Preparation and Characterization of AgNPs in Yerba Extract

The ecofriendly green synthesis of AgNPs was performed using yerba mate extract. The reduction of Ag^+^ ions was confirmed by the apparent color change from straw to dark brown that occurred after 2 h of synthesis (Figure 1A). The formation of AgNPs was confirmed by UV-Vis analysis (Figure 1B). The spectrum of optical absorption regarding metal nanoparticles is dominated by surface plasmon resonances (SPR), which move towards longer wavelengths with increasing particle sizes. The width of each plasmon is related to the nanoparticle size distribution. Spherical particles show only one SPR band in the absorption spectrum, while irregular particles may give rise to two or more SPR bands [34]. In this case, a single peak was observed at a wavelength of 430 nm, which may indicate the formation of spherical nanoparticles. The average particle size of the nanoparticles was measured using the dynamic light scattering technique and was 94.96 ± 3.33 nm (PDI 0.240 ± 0.009 nm). The obtained results show that the yerba mate extract is suitable for the preparation of silver nanoparticles.

In order to recognize the surface charge, the zeta potential of the produced AgNPs was analyzed. Generally, nanoparticles that exhibit zeta potentials below −20 mV are considered unstable, as they cause the particles to precipitate out of the solution. In contrast, nanoparticles with a zeta potentials greater than 20 mV are considered stable [34]. In this study, the zeta potential value of AgNPs was approximately −27 mV, which indicates high stability of the AgNP extract.

### 3.2. Preparation and Characterization of Active Double-Layered Films

The next step in this study was the incorporation of three active ingredients into the 1st layer (FUR) of the double-layered films (FUR/CHIT+HGEL): ethanol extract of curcumin enriched with rosemary essential oil, montmorillonite, and yerba mate extract with AgNPs. Each of the ingredients enriched the film with specific properties.

Two layers are clearly visible in the cross-section of the SEM film photomicrographs (Figure 1C). In addition, after adding yerba mate extract with nanoparticles, silver nanoparticles were clearly visible in the SEM photomicrographs (Figure 1D).

The optical properties of the film were determined by measuring the absorbance within the range of 300–600 nm (Figure 1E). The composite films showed a deep peak of absorbance at 430 nm, which can be attributed to the presence of AgNPs [34] and curcumin extract [33]. This type of packaging can prevent lipid oxidation since the addition of the active ingredients to the film enhances the UV-barrier properties of the materials. The reason for such behavior is the presence of phenolic compounds in curcumin [35] and plant extracts with AgNPs [36], which are responsible for absorbing UV radiation.

### 3.3. Structural Characterization of Active Double-Layered Films

In Figure 2, a topographic 3D image is shown of the nanocomposite film surface imaged with an atomic force microscope (AFM). Pure furcellaran films have a dense and nonporous structure, which has been visualized by both the SEM and AFM methods [5]. The morphology of pure chitosan films is characterized by significant roughness [37], which may be due to the presence of crystalline domains. The creation of double-layered films completely changes the surface morphology of such films. The SEM and AFM photomicrographs taken from the FUR side show that the structure is evenly rough (Figure 2D). As the concentration of active substances increased, the roughness and porosity of the film also increased (Figure 2A–C). In this study, it is probable that furcellaran bound to nanoparticles and curcumin ethanol extract and compressed the polymer structure at these points. The structure of the furcellaran film with AgNPs from the yerba mate extract is shown in Figure 2F and an increase in the roughness of the film can be clearly observed. The addition of curcumin also affects the roughness of the films (Figure 2E). The FUR+curcumin ethanol extract enriched with rosemary oil made the furcellaran film more coarse. This shows that the loaded curcumin increases the roughness of the film [38]. The presence of MMT in the nanocomposite may also affect the roughness of the surface [39]. In addition, the SEM images show the lack of a homogeneous structure, which may be related to the agglomeration of AgNPs, MMT, and unbound curcumin. Such behavior may contribute to lowering the values of the functional parameters of the film.

### 3.4. Color Parameters of Active Double-Layered Films

The color of the film surface was defined by the CIE L*, a*, and b*-values and is shown in Table 1. The FUR/CHIT+HGEL films were transparent with a yellowish tinge, and the brightness of the film was 85.62. The addition of the active ingredients significantly decreased the brightness value, while the values of a* and b* (indicating greenness–redness and blueness–yellowness, respectively) increased significantly. The color change of the composite films was mainly attributed to the color of AgNPs and the ethanol extract of curcumin. Consequently, the total color difference (ΔE) of the composite films increased significantly in comparison to the control film. This result is consistent with visual observation (Table 1). This type of packaging can easily extend the shelf life of a food product, as the phenolic compounds contained in natural pigments can absorb UV radiation and partially visible light [40].

### 3.5. Water Content, Solubility, WVTR, and WCA of Active Double-Layered Films

The addition of the active ingredients to the FUR layer slightly decreased the WC value (Table 2), which may be related to the hydrophobic nature of the curcumin ethanol extract enriched with rosemary essential oil and montmorillonite. A similar situation was observed for the solubility of the films, which was reduced, but the changes were not statistically significant.

Generally, the dispersion of MMT in a biopolymer matrix should reduce water vapor permeability. However, such a trend was observed only in the case of composite No. 1. In contrast, in the case of composite No. 2 and composite No. 3, the WVTR values either increased or no significant changes were observed. In the case of dispersed systems, the mass transfer takes place through the continuous phase, which occupies most of the volume phase. On the other hand, a very small proportion was occupied by MMT particles, the amount of which was not enough to slow down the transfer of water vapor. Similar conclusions were reached by Jiang, Yin, Zhou, Wang, Zhong, Xia, Deng, and Zhao [39] when preparing films based on chitosan, *Akebia trifoliata* (Thunb.) Koidz. peel extracts, and MMT as well as Alexandre et al. [41], who studied gelatin films fortified with ginger essential oil and MMT. Moreover, plant extracts increase the WVTR value. On the other hand, with the increase in the concentration of the hydrophobic curcumin extract, too much of the ingredient could have occurred, which was associated with a lack of potential sites of interaction.

The hydrophobicity of the film was tested by measuring the water contact angle (WCA) on the surface of the film (Table 2). In order to evaluate the hydrophobic/hydrophilic nature of the films, WCA was tested separately for the FUR and for the CHIT+HGEL layers. The mean WCA value of the FUR layer in the control film was 90.08°, which proves its high hydrophobicity [42]. The introduction of the active ingredients caused a significant decrease in the WCA value (down to 24.40°), resulting in a strong hydrophilic nature of the film. Moreover, the addition of the active ingredients to the FUR layer influenced the WCA values of the CHIT+HGEL layer. The WCA value decreased, which contributed to the deterioration of the film’s hydrophobic nature. The recorded decrease may be due to the porosity and roughness of the obtained composite film. In addition, the surface roughness also affects the hydrophilic nature of the material [41]. Thus, the obtained result is consistent with the analysis of the microstructure of the obtained films (Figure 2). The presented films of a hydrophilic nature should not be used as packaging materials for food products with high humidity.

### 3.6. Thermal Properties of Active Double-Layered Films

The addition of the active ingredients to the 1st layer of the double-layered films caused a significant decrease in the ΔH values, which can be attributed to the deterioration of the film’s thermal properties (Table 3 and Figure 3). Such a phenomenon shows that a small addition of active ingredients can improve thermal properties, while increasing the concentration causes incompatibility of the film components, translating into the deterioration of their thermal properties. The deterioration of the thermal properties of the film may result from the presence of essential oil, which causes a plasticizing effect and, consequently, hinders the crystallization of biopolymers [43].

### 3.7. Mechanical Properties of Active Double-Layered Films

The thickness of the control film was 0.088 mm, and along with the increase in the concentration of additives, the values of this parameter increased (Table 1). This behavior can be attributed to the appearance of more solids [40]. The addition of active ingredients, regardless of their concentration, drastically worsened the mechanical properties of the film (Table 3). Only the elongation at break (EAB) parameter increased, which indicates an increase in the flexibility of the film, resulting from the presence of rosemary oil (Dairi et al., 2019). Rostami and Esfahani [44] received mucilage of *Melissa officinalis* seed gum/montmorillonite-based films and noticed that the addition of MMT to the film increased the values of the mechanical properties to close to the strength values of LDPE films. Jiang, Yin, Zhou, Wang, Zhong, Xia, Deng, and Zhao [39] came to a completely different conclusion when adding MMT to a chitosan film. They observed a significant decrease in the TS and EAB values and explained this with agglomerated nanoparticles. In this work, MMT was selected as the active additive to improve the mechanical properties of the tested films. The obtained results indicate that there are too many active ingredients in 1st layer, which may result in a lack of an ordered structure. A solution to this problem may be to locate a specific active additive in one particular layer. Thus, the active additives will not compete to form interactions with the furcellaran chain.

### 3.8. Antioxidant Activity

Two different methods were used to determine the antioxidant activity of the films: the DPPH and FRAP methods (Figure 4). Overall, from the results, it is possible to clearly observe a tendency towards increasing the antioxidant capacity of the films through the incorporation of bioactive ingredients. By analyzing the FRAP method, a statistically significant increase in the antioxidant activity of the films can be observed as a function of the CUR proportion. However, when evaluating the DPPH method, it was noticed that there was a significant increase in the radical scavenging capacity of the films only for higher concentrations of CUR. These different profiles may be explained by the mechanisms of action in each method. While the FRAP assay measures the reducing capacity upon the reduction of ferric ions, the DPPH method is based on the H-transfer mechanism.

Although other active ingredients may also contribute to the increase in the antioxidant capacity of the films, it is clear that the main reason is the presence of CUR. This molecule has been found to be an effective and safe natural antioxidant in different in vitro assays [45]. CUR has a marked capacity for iron binding, suggesting its main action as a peroxidation inhibitor. It has been reported that compounds with structures containing C–OH and C=O functional groups can chelate metal ions [46], explaining the profile of the antioxidant activity observed in the FRAP assay. Conversely, although hydrogen atom abstraction from the phenolic ring contributes to CUR’s antioxidant activity, its abstraction is a little difficult since CUR’s phenolic hydrogen atoms are intramolecularly H-bonded to the adjacent methoxy groups [47]. Consequently, the reaction of DPPH radicals diminishes via curcumin in alcoholic media, corroborating the differences in the antioxidant activity profiles determined by the two methods.

### 3.9. Antimicrobial Activity

The antibacterial ability of the films against several microorganisms is shown in Table 4. The control films did not show any antimicrobial effect. In the case of composite No. 2, there was a visible antimicrobial effect against *S. aureus*, *E. faecalis*, and *E. coli* and a slight effect against *S. enterica* and *P. aeruginosa*. The lack of any antimicrobial effect of composite No. 3 may be the result of an excessive concentration of active additives, and such behavior indicates that it is very important when designing the package to select the right ingredients and concentrations. Moreover, the obtained films did not show any antifungal activity. The antimicrobial potential of the presented films may result from the presence of curcumin extract and AgNPs in the plant extracts. In previous works, we indicated the high antimicrobial potential of AgNPs in plant extracts [30] and curcumin extract with lemongrass essential oil [48]. Packaging that shows antimicrobial activity can be used as a material for extending the shelf life of food products.

### 3.10. pH-Responsive Color Change

Changes in pH values are one of the commonly used chemical markers indicating the growth of bacteria [44]. Therefore it was decided to test the films for their intelligent properties. The visual change in the color of the composite films is shown in Figure 5. In acidic solutions, the films had an enhanced yellow-orange color, while in an alkaline environment the color changed to an intense red. The reason for this behavior is that some functional groups of curcumin are deprotonated at an alkaline pH, making the molecule more polar. This result could indicate a potential use for detecting pH changes in packaged food [44]. However, further research is necessary, which will concern the behavior of the indicators during the storage of model food products in vivo.

## 4. Conclusions

For the first time, a double-layered film based on furcellaran, chitosan, and gelatin hydrolysate and three active ingredients (curcumin, MMT, and AgNPs in yerba mate extract) were successfully obtained. Due to the presence of curcumin extract, the obtained films had a yellow color and were characterized by a barrier against UV radiation. Moreover, in an in vitro test, the presented films changed color depending on the pH environment, which may indicate their potential as indicators of the freshness of food products. The decrease in the value of the mechanical and thermal properties evidently proves that there are too many active ingredients in the 1st film layer. It is necessary to refine the number of layers as well as to clarify the location of a specific active ingredient in a specific layer. However, with the composition of the active ingredients in composite No. 1, it was possible to lower the WVTR parameter. The obtained results prove the high application potential of such packaging materials due to their active (antimicrobial and antioxidant) and intelligent properties. However, it is necessary to refine the active layers of the film. In addition, further research is needed to determine the migration of AgNPs to food products during their storage.

## Figures and Tables

**Figure 1 polymers-14-04283-f001:**
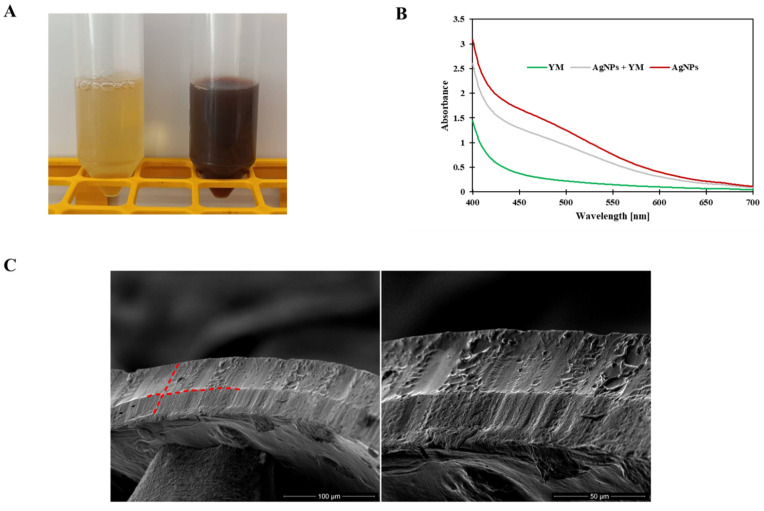
(**A**) The appearance of solutions without AgNPs (on the left) and with AgNPs (on the right); (**B**) UV-Vis spectra of solutions with and without AgNPs; (**C**) cross-section of double-layered films; (**D**) surface of double-layered films (red circles indicate silver nanoparticles); (**E**) UV-Vis spectra of composite films.

**Figure 2 polymers-14-04283-f002:**
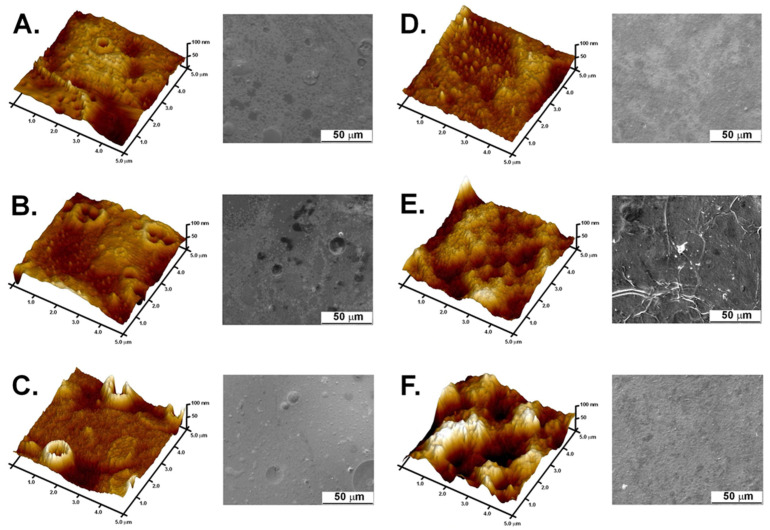
SEM and AFM images of: (**A**) composite No. 1; (**B**) composite No. 2; (**C**) composite No. 3; (**D**) FUR/CHIT+HGEL; (**E**) FUR+CUR-REO; and (**F**) FUR+AgNPs-YM.

**Figure 3 polymers-14-04283-f003:**
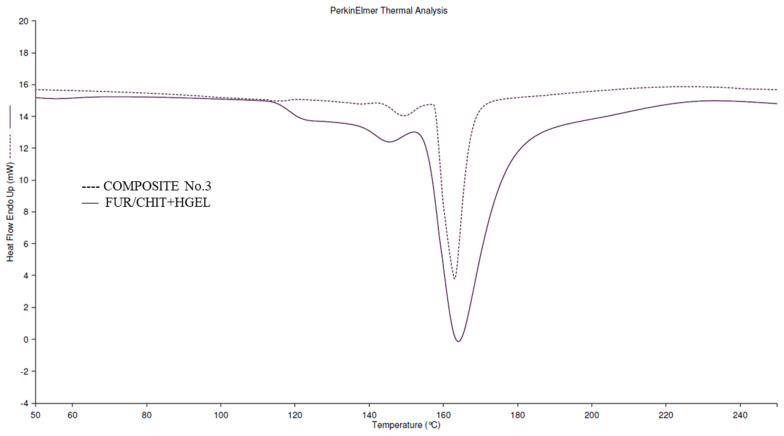
Typical DSC curves of tested films.

**Figure 4 polymers-14-04283-f004:**
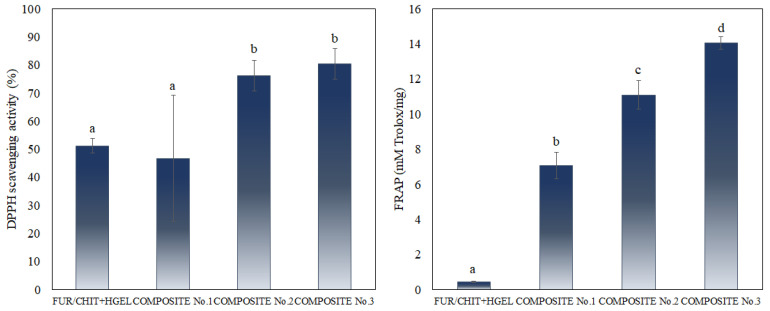
Antioxidant activity of active double-layered films. Different lettering indicates significant differences (*p* < 0.05).

**Figure 5 polymers-14-04283-f005:**
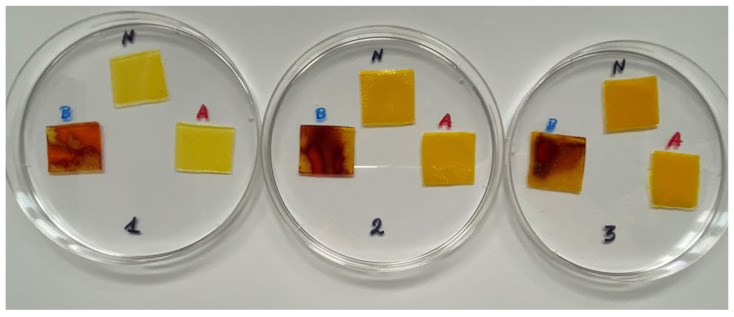
The pH responses of the composite films in contact with acid (A), neutral (N), and alkali (B) conditions. Abbreviations: (1) Composite No. 1; (2) Composite No. 2; (3) Composite No. 3.

**Table 1 polymers-14-04283-t001:** Color parameters of active double-layered films.

Parameter	FUR/CHIT+HGEL	Composite No. 1	Composite No. 2	Composite No. 3
L*	85.62 ^d^ ± 0.73	78.12 ^c^ ± 0.58	74.49 ^b^ ± 0.32	73.39 ^a^ ± 0.23
a*	−0.82 ^a^ ± 0.16	3.90 ^b^ ± 0.27	8.59 ^c^ ± 0.43	9.16 ^d^ ± 0.37
b*	29.11 ^a^ ± 2.35	76.28 ^b^ ± 0.49	80.20 ^c^ ± 0.64	79.37 ^c^ ± 0.41
ΔE	-	48.00	53.13	52.68
Appearance	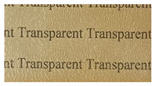	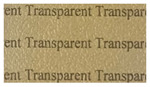	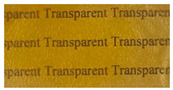	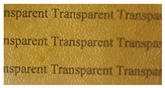

* Values are expressed as means ± SD. Different lettering ^a–d^ in the same row indicates significant differences (*p* < 0.05).

**Table 2 polymers-14-04283-t002:** Water content, solubility, WVTR, and WCA of active double-layered films.

Properties *	FUR/CHIT+HGEL	Composite No. 1	Composite No. 2	Composite No. 3
Thickness (mm)	0.088 ^a^ ± 0.004	0.110 ^b^ ± 0.006	0.120 ^c^ ± 0.006	0.132 ^d^ ± 0.007
Water content (%)	22.42 ^b^ ± 3.08	18.94 ^a^ ± 1.79	16.64 ^a^ ± 1.50	18.54 ^a^ ± 0.42
Solubility (%)	67.74 ^a^ ± 1.25	61.07 ^a^ ± 1.53	59.41 ^a^ ± 9.35	62.99 ^a^ ± 3.79
WVTR (g m^−2^ d^−1^)	739.77 ^b^ ± 10.65	724.05 ^a^ ± 3.73	741.01 ^b^ ± 10.81	735.37 ^ab^ ± 4.50
WCA	CHIT+HGEL	CHIT+HGEL	CHIT+HGEL	CHIT+HGEL
	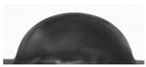 92.14 ^c^ ± 1.43	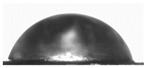 79.30 ^a^ ± 1.34	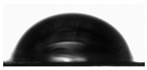 80.29 ^ab^ ± 1.96	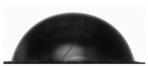 80.95 ^b^ ± 0.70
	FUR	FUR+MMT+AgNPs+CUR	FUR+MMT+AgNPs+CUR	FUR+MMT+AgNPs+CUR
	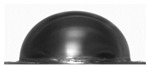 90.08 ^b^ ± 0.48	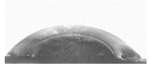 24.40 ^a^ ± 1.40	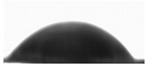 25.38 ^a^ ± 1.17	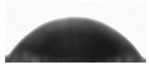 24.48 ^a^ ± 1.23

* Values are expressed as means ± SD. Different lettering ^a–d^ in the same row indicates significant differences (*p* < 0.05).

**Table 3 polymers-14-04283-t003:** Thermal and mechanical properties of active double-layered films.

Properties *	FUR/CHIT+HGEL	Composite No. 1	Composite No. 2	Composite No. 3
T_m_ (°C)	160.7 ^b^ ± 6.3	156.9 ^ab^ ± 2.8	151.1 ^a^ ± 3.9	164.1 ^b^ ± 1.3
ΔH (J/g)	221.43 ^bc^ ± 3.09	228.57 ^c^ ± 15.24	201.10 ^b^ ± 14.36	109.27 ^a^ ± 15.62
Max breaking load (N)	14.75 ^b^ ± 1.04	6.68 ^a^ ± 1.42	6.51 ^a^ ± 1.23	5.37 ^a^ ± 0.98
Tensile strength (kN/m)	0.97 ^b^ ± 0.07	0.45 ^a^ ± 0.09	0.43 ^a^ ± 0.08	0.36 ^a^ ± 0.07
Elongation at break (%)	16.24 ^a^ ± 1.83	17.70 ^a^ ± 3.26	21.10 ^b^ ± 4.05	15.45 ^a^ ± 0.98
Modulus of elasticity (MPa)	209.90 ^b^ ± 27.49	63.59 ^a^ ± 14.10	77.55 ^a^ ± 12.06	57.04 ^a^ ± 13.58

* Values are expressed as means ± SD. Different lettering ^a–c^ in the same row indicates significant differences (*p* < 0.05).

**Table 4 polymers-14-04283-t004:** Antimicrobial activity of tested films.

Microbial Strain	Sample			
	FUR/CHIT+HGEL	Composite No.1	Composite No.2	Composite No.3
*Staphylococcus aureus* ATCC 29213	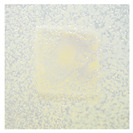 no antimicrobial effect	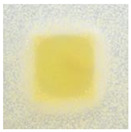 moderate antimicrobial effect and bacterial growth only on the film border	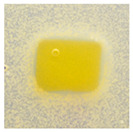 good antimicrobial effect	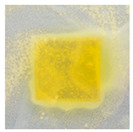 no antimicrobial effect alteration of film and bacterial growth
*Enterococcus faecalis* ATCC 29212	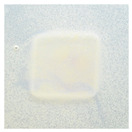 no antimicrobial effect	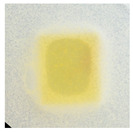 moderate antimicrobial effect and bacterial growth only on the film border	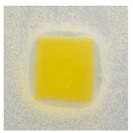 good antimicrobial effect—no bacterial growth and a narrow inhibition zone	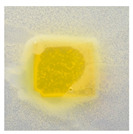 no antimicrobial effect alteration of film and bacterial growth
*Escherichia coli* ATCC 25922	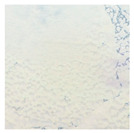 no antimicrobial effect	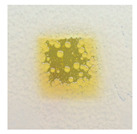 partial effect—no microbial growth in the center of the sample	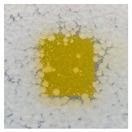 moderate antimicrobial effect—few colonies grew over the sample	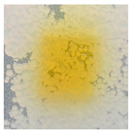 no antimicrobial effect
*Salmonella enterica* ATCC BAA664	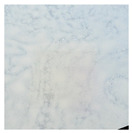 no antimicrobial effect	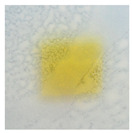 no antimicrobial effect	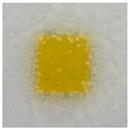 moderate antimicrobial effect—few colonies grew over the sample	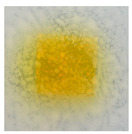 no antimicrobial effect
*Pseudomonas aeruginosa* ATCC 9027	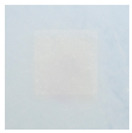 no antimicrobial effect	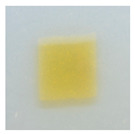 no antimicrobial effect	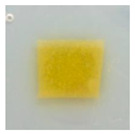 some antimicrobial effect—lower bacterial growth at the center of the film	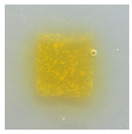 no antimicrobial effect; alteration of growth and film
*Candida krusei* ATCC 6258	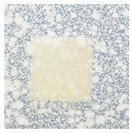 no antimicrobial effect	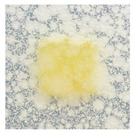 no antimicrobial effect	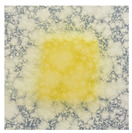 no antimicrobial effect	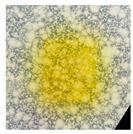 no antimicrobial effect
*Candida albicans* ATCC 90028	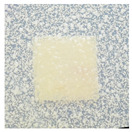 no antimicrobial effect	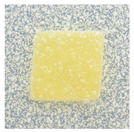 no antimicrobial effect	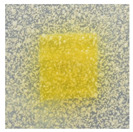 no antimicrobial effect	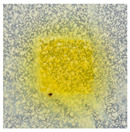 no antimicrobial effect
*Aspergillus niger* ATCC 16404	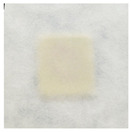 no antimicrobial effect	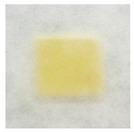 no antimicrobial effect	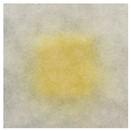 no antimicrobial effect	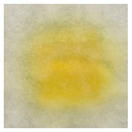 no antimicrobial effect
*Aspergillus flavus* ATCC 204304	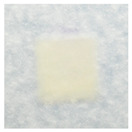 no antimicrobial effect	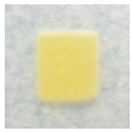 no antimicrobial effect	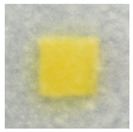 no antimicrobial effect	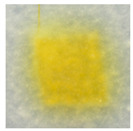 no antimicrobial effect

## Data Availability

Not applicable.

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
