# Peer review of "Double-Layered Films Based on Furcellaran, Chitosan, and Gelatin Hydrolysates Enriched with AgNPs in Yerba Mate Extract, Montmorillonite, and Curcumin with Rosemary Essential Oil"

_polymers, 2022, doi:10.3390/polym14204283_

Round 1

Reviewer 1 Report

The present article investigated the development of double-layered films based on furcellaran, chitosan and gelatin containing AgNPs, Montmorillonite, and Curcumin Closed in Rosemary EO for active food packaging applications. This research has certain scientific significance. I have the following comments.

 Abstract

- Include the most important results.

- The abstract should be revised and restructured.

 - The objectives of the study should be explained more precisely.

 Materials and methods

- The description in some part of the materials and method was ambiguous and not clear.

- Section 2.2, 2.4 (method of extraction), 2.5., 2.11, 2.12, 2.13, and 2.17 should be explained completely.

- Section 2.7 is confusing. It must be rewritten. What was the final concentration of each compound in the prepared film? The proportions used must be specified precisely (w/v, w/w etc.).

- What was the concentration of the essential oil used in the composition of the film?

- If the antimicrobial properties of the prepared film have been determined, it should be provided.

Results and discussion

- In general, the discussion part of the article is relatively weak.

-Please strengthen the discussion of the manuscript in the required sections such as 3.1, 3.3, 3.4, and 3.5.

-Conclusion has been described very poorly. It should be improved.

With the best,

Author Response

Reviewer 1:

Comments and Suggestions for Authors

The present article investigated the development of double-layered films based on furcellaran, chitosan and gelatin containing AgNPs, Montmorillonite, and Curcumin Closed in Rosemary EO for active food packaging applications. This research has certain scientific significance. I have the following comments.

 Abstract

Comment: - Include the most important results. The abstract should be revised and restructured.

Response: The information have been added.

Comment:  - The objectives of the study should be explained more precisely.

Response: We have re-described the purpose of the work in the Introduction section.

 Materials and methods

Comment: - The description in some part of the materials and method was ambiguous and not clear.

Response: We have reorganized this Section and add every information about the specific procedures.

Comment: - Section 2.2, 2.4 (method of extraction), 2.5., 2.11, 2.12, 2.13, and 2.17 should be explained completely.

Response: We have added more information.

Comment: - Section 2.7 is confusing. It must be rewritten. What was the final concentration of each compound in the prepared film? The proportions used must be specified precisely (w/v, w/w etc.).

Response: The corrections have been made.

Comment: - What was the concentration of the essential oil used in the composition of the film?

Results: The concentration of the essential oil in the film-forming solution based on furcellaran (1st layer) of Composite No. 1 is 0.09375% v/v, in Composite No. 2 it is 0.1875% v/v, and in Composite No. 3 it is 0.25% v/v.

Comment: - If the antimicrobial properties of the prepared film have been determined, it should be provided.

Response: We have added this method.

Results and discussion

Comment: - In general, the discussion part of the article is relatively weak.

Response: We have added more information and methods.

Comment: -Please strengthen the discussion of the manuscript in the required sections such as 3.1, 3.3, 3.4, and 3.5.

Response: We have added more information in every section.

Comment: -Conclusion has been described very poorly. It should be improved.

Response: We have added more information.

Reviewer 2 Report

The reviewed manuscript presents the results of work on a two-layer film produced with the use of furellaran, chitosan and gelatin, modified with curcumin and biosynthesized AgNP. the work is innovative and interesting and the manuscript is well written. However, as a reviewer, I have a few comments:
Have the types and proportion of curcuminoids contained in curcumin used been determined?
What was the total thickness of the different types of films obtained?
What is the size of AgNPs?
 After what time was the antioxidant activity by DPPH method analyzed?
Can the plasticizing effect be additionally caused by the presence of TWEEN 80 dispersant?
Are there any solvent residues in the resulting foils?
For better visualization of the analyzed effects, the obtained DSC curves might be included.

Author Response

Reviewer 2:

Comments and Suggestions for Authors

The reviewed manuscript presents the results of work on a two-layer film produced with the use of furellaran, chitosan and gelatin, modified with curcumin and biosynthesized AgNP. the work is innovative and interesting and the manuscript is well written. However, as a reviewer, I have a few comments:

Comment: Have the types and proportion of curcuminoids contained in curcumin used been determined?

Response: Thank you for your question. This work was aimed at the selection of active ingredients, after many analyzes we are aware that the next step will be to clarify the new composition of double-layered films. At the moment, the exact composition of the curcumin extract has not been determined due to the inability to conduct such a study. However, as part of the project, the composition of the extract will be analyzed (now we are looking for a possibility to perform such an analysis), as well as a specific and global migration test for the food simulants model.

Comment: What was the total thickness of the different types of films obtained?

Response: We have added this information into Table 1.

Comment: What is the size of AgNPs?

Response: The particle size is on the limit: 94.96±3.33 nm. We have added this information to Manuscript.

Comment: After what time was the antioxidant activity by DPPH method analyzed?

Response: It was measured after 30 minutes. This information has been added to the methodology section.

Comment: Can the plasticizing effect be additionally caused by the presence of TWEEN 80 dispersant?

Response: First of all, TWEEN 80 works by stabilizing the CUR-REO extract. However, together with rosemary oil and glycerol, it can contribute to making the material more plasticized.

Comment: Are there any solvent residues in the resulting foils?

Response: Ethanol evaporates faster than water, but it can hydrogen bond with both water and polysaccharide, so theoretically it could have stayed a bit.

Comment: For better visualization of the analyzed effects, the obtained DSC curves might be included.

Response: We have added typically curves of FUR/CHIT+HGEL and COMPOSITE No.3 for better visualization. We only added two, because if we add all of them, it's hard to read the chart.

Reviewer 3 Report

Comments

Summary

This article has described the fabrication of “Double-layered Films Based on Furcellaran, Chitosan, Gelatin Hydrolysates Enriched with AgNPs in Yerba Mate Extract, Montmorillonite and Curcumin Closed in Rosemary Essential Oil”. For last many decades, different synthetic polymers have been used as a packaging material because of their lightness, softness, and transparency. However, the non-biodegradability of synthetic polymers is the leading cause of serious ecological problems and that is why use of biodegradable polymers are important in the packaging industry. Currently, possibilities of incorporation of biopolymers in the smart food packaging industry are undoubtedly a topic of interest for researchers and use of bioplastic in packaging industry should be the probable future.

Considering the significance of bioplastic in the packaging industry, this topic of research undeniably has an importance in the scientific community. However, based on the complicated composition of the fabricated bi-layer films and insufficient characterizations of them to support the conclusion, I do not think this manuscript is suitable for publication in “Polymers”. Some of my major concerns are enlisted below-

Major Comments

1.       Authors have mentioned in Line No- 324, “The obtained results indicate that there are too many active ingredients in 1 layer, which may result in a lack of ordered structure”. I have the exact same concern for this work too. Authors have used too many components such as Furcellaran, Chitosan, Gelatin Hydrolysates Enriched with AgNPs in Yerba Mate Extract, Montmorillonite and Curcumin Closed in Rosemary Essential Oil and for this the exact conclusion of this work is very difficult to understand.

2.       Both Ag NPs and curcumin have antioxidant property and antimicrobial property. Even Chitosan, Gelatin Hydrolysates, and Rosemary Essential Oil all have antimicrobial property. It is very painful to understand why Authors have used too many ingredients to get a single property.

3.       It is very unclear why Authors have used Yerba Mate Extract as a reducing agent? There are many bio-extracts are available those can also be engaged for the bio-reduction process.

4.       Explanation of WVTR and WCA results are very confusing and hard to understand. For better understanding of the bulk property of nanocomposite films, Authors may consider to conduct TEM analysis. TEM study will make it clear whether clay particles are in the intercalated state or in the exfoliated state. If MMT clay particles are in the exfoliated state, it should improve the WVTR by forming the tortuous path.

5.       Thermal property of the composite film could be better understood by the thermogravimetric analysis (TGA).

6.        To claim any film to be used as a potential candidate for packaging material, mechanical properties should be as good as the conventional packaging polymers or at least nearby. This fabricated film does not meet the necessary mechanical properties.

7.       There are few reports available based on Furcellaran, Gelatin Hydrolysates, and nanoparticles reinforced composite films and their applications as a potential packaging material. References are mentioned below-

·         Tkaczewska, J., Kulawik, P., Jamróz, E., Guzik, P., ZajÄ…c, M., Szymkowiak, A., & Turek, K. (2021). One-and double-layered furcellaran/carp skin gelatin hydrolysate film system with antioxidant peptide as an innovative packaging for perishable foods products. Food Chemistry351, 129347.

·         Jancikova, S., Jamróz, E., Kulawik, P., Tkaczewska, J., & Dordevic, D. (2019). Furcellaran/gelatin hydrolysate/rosemary extract composite films as active and intelligent packaging materials. International Journal of Biological Macromolecules131, 19-28.

·         Jamróz, E., Kopel, P., Juszczak, L., Kawecka, A., Bytesnikova, Z., Milosavljevic, V., & Makarewicz, M. (2019). Development of furcellaran-gelatin films with Se-AgNPs as an active packaging system for extension of mini kiwi shelf life. Food Packaging and Shelf Life21, 100339.

Authors should include the importance and uniqueness of this manuscript compared to the existing literature.

8.       For a packaging material, it should satisfy three important properties such as barrier properties, chemical resistant properties, and mechanical properties. Authors should include the data related to chemical resistant properties of the fabricated film.

Minor Comments

1.    Authors should include few sentences while explaining synthesis process and characteristic properties instead of only mentioning references of their previous work. By doing this the manuscript will be easier to understand for the reader.

Author Response

Reviewer 3:

Summary

This article has described the fabrication of “Double-layered Films Based on Furcellaran, Chitosan, Gelatin Hydrolysates Enriched with AgNPs in Yerba Mate Extract, Montmorillonite and Curcumin Closed in Rosemary Essential Oil”. For last many decades, different synthetic polymers have been used as a packaging material because of their lightness, softness, and transparency. However, the non-biodegradability of synthetic polymers is the leading cause of serious ecological problems and that is why use of biodegradable polymers are important in the packaging industry. Currently, possibilities of incorporation of biopolymers in the smart food packaging industry are undoubtedly a topic of interest for researchers and use of bioplastic in packaging industry should be the probable future.

Considering the significance of bioplastic in the packaging industry, this topic of research undeniably has an importance in the scientific community. However, based on the complicated composition of the fabricated bi-layer films and insufficient characterizations of them to support the conclusion, I do not think this manuscript is suitable for publication in “Polymers”. Some of my major concerns are enlisted below-

Major Comments

  1. Comment: Authors have mentioned in Line No- 324, “The obtained results indicate that there are too many active ingredients in 1 layer, which may result in a lack of ordered structure”. I have the exact same concern for this work too. Authors have used too many components such as Furcellaran, Chitosan, Gelatin Hydrolysates Enriched with AgNPs in Yerba Mate Extract, Montmorillonite and Curcumin Closed in Rosemary Essential Oil and for this the exact conclusion of this work is very difficult to understand.
  2. Comment: Both Ag NPs and curcumin have antioxidant property and antimicrobial property. Even Chitosan, Gelatin Hydrolysates, and Rosemary Essential Oil all have antimicrobial property. It is very painful to understand why Authors have used too many ingredients to get a single property.

Response: We decided to combine Reviewer's Comment 1 and Comment 2 and gave one answer. Our research team has developed many types of films based on furcelleran, chitosan and hydrolysate. We know the disadvantages of these films and we wanted to reduce their limits.

Thank you very much for the tip, now we can see that it is actually necessary to refine the justification for the use of the mentioned active ingredients. The plan was as follows, we wanted to present a biopolymer matrix in a completely different way. We have already developed a matrix based on furcellaran and chitosan [1] as well as a matrix based on furcellaran-chitosan and gelatin hydrolysate [2]. Unfortunately, these solutions did not improve the functional properties of single-layered films. We decided to design the presented material in a completely different way, namely in the form of double-layered films. The gelatin hydrolysate itself does not form a film, but has antioxidant properties and gives the film great flexibility, hence these two components (chitosan as biopolymer matrix and gelatin hydrosylate as active ingredient) formed one of the layers. On the other hand, the matrix in the second biopolymer layer was furcellaran. We have been working on these biopolymers for many years and unfortunately it has too high solubility, poor mechanical properties and no biological properties.

Therefore, our main goal was to improve this furcellaran-based layer. The addition of MMT was intended to increase stiffness and to reduce the WVTR parameter which is crucial when using packaging materials for food products. Then, the use of AgNPs obtained by the ex situ method was also aimed at improving the antioxidant and antimicrobial properties of the tested films. The last step was to add curcumin extract to impart active and intelligent properties. And here we wanted to fine-tune the most optimal ratio by adding the extract in different concentrations. Moreover, we wanted CUR and MMT to reduce the solubility of these films as furcellaran film is 100% soluble. In the current manuscript, we have added a method to confirm the intelligent properties that were caused by the presence of curcumin.

In the first place, we tried to add curcumin extract and AgNPs extract to the chitosan layer, but here we had no compatibility (very bad dispersion of ingredients in the biopolymer matrix, manifested e.g. by uneven color) (it was our preliminary study). Our goal was not to incorporate as many active ingredients as possible into the film so that the films have an antimicrobial effect. Our goal was to improve the specific functional properties of the film by adding a specific component. We decided that we would try to add all the ingredients to the furcellaran layer, because unfortunately this layer needed the most improvement. What else was an additional advantage is that there are no studies showing the addition of CUR, MMT or AgNPs in plant extracts to biopolymer films.

Additionally, the arrangement of double-layered films in the form of FUR/CHIT+HGEL was also not accidental. We were not able to obtain a film, eg CHIT/FUR+HGEL, due to the fact that the chitosan solution does not take the form of a gel throughout the drying process, so it would not be possible to apply another biopolymer layer.

Moreover, we noticed that there are works in the literature that also concern the modification of biopolymer films with several types of active ingredients [3].

In fact, reading the introduction, you may not understand what the Authors wanted to do, now we have re-edited the entire introduction for a better understanding of the work. Thank You for this comment, now we see our bad explanation in Introduction.

  1. Comment: It is very unclear why Authors have used Yerba Mate Extract as a reducing agent? There are many bio-extracts are available those can also be engaged for the bio-reduction process.

Response: We are experimenting with obtaining AgNPs in various plant extracts. We have published a paper where we use two different plant extracts of AgNPs as an additive to double-layer films [4].

We chose yerba mate extract because we had already worked with this extract as an additive to the films [5]. Now we wanted to use this as a reducing agent for AgNPs. In addition, during the synthesis of AgNPs in yerba mate, we have high reproducibility of results and a fast time to obtain nanoparticles. That is why we decided to use this extract in the preparation of nanoparticles.

  1. Comment: Explanation of WVTR and WCA results are very confusing and hard to understand. For better understanding of the bulk property of nanocomposite films, Authors may consider to conduct TEM analysis. TEM study will make it clear whether clay particles are in the intercalated state or in the exfoliated state. If MMT clay particles are in the exfoliated state, it should improve the WVTR by forming the tortuous path.

Response: We would like to thank you for your suggestion. We have reworked this section. In our opinion, the AFM and SEM analyzes sufficiently show the structure of the presented films. At the moment, we do not have the possibility to perform a TEM analysis, of course, as soon as there is an opportunity, we will conduct this  study.

  1. Comment: Thermal property of the composite film could be better understood by the thermogravimetric analysis (TGA).

       Response: Thank you for your comment, however, we are unable to perform TGA testing. We only had DSC equipment at our disposal, which also used to characterize the thermal properties of the film.

  1. Comment: To claim any film to be used as a potential candidate for packaging material, mechanical properties should be as good as the conventional packaging polymers or at least nearby. This fabricated film does not meet the necessary mechanical properties.

       Response: Here, too, we agree with this remark. Our work consists in improving these properties, therefore the aim of this work was to add MMT and AgNPs in yerba extract as fillers that will not only improve the mechanical properties but also the water vapor barrier properties. Unfortunately, it was not possible to do so. This work is also a response to the fact that adding so many ingredients will definitely not contribute to the implementation of this type of film on the market.

  1. Comment: There are few reports available based on Furcellaran, Gelatin Hydrolysates, and nanoparticles reinforced composite films and their applications as a potential packaging material. References are mentioned below-

  • Tkaczewska, J., Kulawik, P., Jamróz, E., Guzik, P., ZajÄ…c, M., Szymkowiak, A., & Turek, K. (2021). One-and double-layered furcellaran/carp skin gelatin hydrolysate film system with antioxidant peptide as an innovative packaging for perishable foods products. Food Chemistry351, 129347.
  • Jancikova, S., Jamróz, E., Kulawik, P., Tkaczewska, J., & Dordevic, D. (2019). Furcellaran/gelatin hydrolysate/rosemary extract composite films as active and intelligent packaging materials. International Journal of Biological Macromolecules131, 19-28.
  • Jamróz, E., Kopel, P., Juszczak, L., Kawecka, A., Bytesnikova, Z., Milosavljevic, V., & Makarewicz, M. (2019). Development of furcellaran-gelatin films with Se-AgNPs as an active packaging system for extension of mini kiwi shelf life. Food Packaging and Shelf Life21, 100339.

Authors should include the importance and uniqueness of this manuscript compared to the existing literature.

       Response: We have add more information to Introduction Section

  1. Comment: For a packaging material, it should satisfy three important properties such as barrier properties, chemical resistant properties, and mechanical properties. Authors should include the data related to chemical resistant properties of the fabricated film.

      Response: In this study, double-layered films were characterized in terms of many parameters, including mechanical properties and barrier properties (UV and WVTR). Moreover, the water resistant properties of the obtained films were determined. The designed experiment gave a negative result and showed that too much active ingredients in the presented biopolymer matrix does not improve the functional properties of the film. This points to the further search for active ingredients that can improve the properties of biopolymer films. In addition, the chemical resistance for this type of material is of secondary importance. The most important functional parameters have been characterized and indicated the negative effect of the addition of such a large amount of active ingredients.

Minor Comments

Comment: 1.    Authors should include few sentences while explaining synthesis process and characteristic properties instead of only mentioning references of their previous work. By doing this the manuscript will be easier to understand for the reader.

Response: We have added more information in Method Section.

  1. Jamróz, E., et al., Composite Biopolymer Films Based on a Polyelectrolyte Complex of Furcellaran and Chitosan. Carbohydrate Polymers, 2021: p. 118627.
  2. Janik, M., et al., Utilisation of Carp Skin Post-Production Waste in Binary Films Based on Furcellaran and Chitosan to Obtain Packaging Materials for Storing Blueberries. Materials, 2021. 14(24): p. 7848.
  3. Zehra, A., et al., Preparation of a biodegradable chitosan packaging film based on zinc oxide, calcium chloride, nano clay and poly ethylene glycol incorporated with thyme oil for shelf-life prolongation of sweet cherry. International Journal of Biological Macromolecules, 2022. 217: p. 572-582.
  4. Jamróz, E., et al., Active Double-Layered Films Enriched with AgNPs in Great Water Dock Root and Pu-Erh Extracts. Materials, 2021. 14(22): p. 6925.
  5. Pluta-Kubica, A., et al., Active edible furcellaran/whey protein films with yerba mate and white tea extracts: Preparation, characterization and its application to fresh soft rennet-curd cheese. International Journal of Biological Macromolecules, 2019.

Round 2

Reviewer 1 Report

-

Reviewer 3 Report

Authors have included results of some newly conducted studies in their revised manuscript. However, I still believe that the design of fabrication, selection of ingredients, and necessary experiments need to be improved to conclude the designed target. Considering the complexity of this work, I am unable to accept this manuscript. Best of luck.